# Progression of Interstitial Fibrosis and Tubular Atrophy in Low Immunological Risk Renal Transplants Monitored by Sequential Surveillance Biopsies: The Influence of TAC Exposure and Metabolism

**DOI:** 10.3390/jcm10010141

**Published:** 2021-01-04

**Authors:** Betty Chamoun, Irina B. Torres, Alejandra Gabaldón, Joana Sellarés, Manel Perelló, Eva Castellá, Xavier Guri, Maite Salcedo, Nestor G. Toapanta, Ignacio Cidraque, Francesc Moreso, Daniel Seron

**Affiliations:** 1Nephrology Departments, Hospital Universitari Vall d’Hebron, 08035 Barcelona, Spain; md.chamounbetty@gmail.com (B.C.); ibtorres@vhebron.net (I.B.T.); jsellares@vhebron.net (J.S.); mperello@vhebron.net (M.P.); ntoapanta@vhebron.net (N.G.T.); ignacio.cidraque@vhir.org (I.C.); dseron@vhebron.net (D.S.); 2Pathology Departments, Hospital Universitari Vall d’Hebron, 08035 Barcelona, Spain; agabaldon@vhebron.net (A.G.); mtsalced@vhebron.net (M.S.); 3Radiology Departments, Hospital Universitari Vall d’Hebron, 08035 Barcelona, Spain; ecastell@vhebron.net (E.C.); xguri@vhebron.net (X.G.); 4Department of Medicine, Autonomous University of Barcelona, 08035 Barcelona, Spain

**Keywords:** tacrolimus, renal transplantation, protocol biopsies, concentration dose ratio, time in therapeutic range, coefficient of variation

## Abstract

The combination of tacrolimus (TAC) and mycophenolate is the most widely employed maintenance immunosuppression in renal transplants. Different surrogates of tacrolimus exposure or metabolism such as tacrolimus trough levels (TAC-C_0_), coefficient of variation of tacrolimus (CV-TAC-C_0_), time in therapeutic range (TTR), and tacrolimus concentration dose ratio (C/D) have been associated with graft outcomes. We explore in a cohort of low immunological risk renal transplants (*n* = 85) treated with TAC, mycophenolate mofetil (MMF), and steroids and then monitored by paired surveillance biopsies the association between histological lesions and TAC-C_0_ at the time of biopsy as well as CV-TAC-C_0_, TTR, and C/D during follow up. Interstitial inflammation (i-Banff score ≥ 1) in the first surveillance biopsy was associated with TAC-C_0_ (odds ratio (OR): 0.69, 95% confidence interval (CI): 0.50–0.96; *p* = 0.027). In the second surveillance biopsy, inflammation was associated with time below the therapeutic range (OR: 1.05 and 95% CI: 1.01–1.10; *p* = 0.023). Interstitial inflammation in scarred areas (i-IFTA score ≥ 1) was not associated with surrogates of TAC exposure/metabolism. Progression of interstitial fibrosis/tubular atrophy (IF/TA) was observed in 35 cases (41.2%). Multivariate regression logistic analysis showed that mean C/D (OR: 0.48; 95% CI: 0.25–0.92; *p* = 0.026) and IF/TA in the first biopsy (OR: 0.43, 95% CI: 0.24–0.77, *p* = 0.005) were associated with IF/TA progression between biopsies. A low C/D ratio is associated with IF/TA progression, suggesting that TAC nephrotoxicity may contribute to fibrosis progression in well immunosuppressed patients. Our data support that TAC exposure is associated with inflammation in healthy kidney areas but not in scarred tissue.

## 1. Introduction

Renal transplantation is the best treatment for end-stage renal disease, since it is associated with a better long-term patient survival and a higher quality of life at a lower cost than dialysis techniques [1]. Since the beginning of the present century and following international guidelines (Kidney Disease: Improving Global Outcomes; KDIGO) maintenance immunosuppression in most renal transplant units is based on the combination of tacrolimus (TAC) and mycophenolate mofetil (MMF) either with or without low-dose steroids [2]. One important limitation for this strategy is that TAC is a drug with a narrow therapeutic window, and the optimal whole blood target levels (TAC-C_0_) during follow up have not been properly defined. Low TAC exposure during the first year has been associated with a higher risk of clinical and subclinical acute rejection and a higher risk of the development of HLA donor-specific antibodies (DSA) in a prospective randomized clinical trial [3]. Additionally, registry studies have shown an association between low TAC exposure and poorer long-term graft survival [4]. Meanwhile, high TAC exposure has been associated with nephrotoxicity, viral infections, and cancer among other toxicities [5,6]. Additionally, tacrolimus is a drug with a low bioavailability (20–30%), and its metabolism rate is related with different genes [7].

To analyze the complex relationship between TAC exposure and clinical outcomes, different parameters can be employed. During the last years, a high intra-patient variability of tacrolimus trough levels during the first years after transplant has been associated with acute rejection, development of de novo DSA, chronic active antibody-mediated rejection, and poorer long-term graft survival [8,9]. It was sustained that patients with high intra-patient variability are outside the therapeutic range during longer periods of time [10,11], and this fact may be especially harmful for those kidneys with a higher HLA incompatibility [12].

In the other hand, to explore the association between tacrolimus metabolism and clinical outcomes, different pharmacokinetic and pharmacodynamic models have been developed, taking into consideration clinical variables such as hematocrit, serum albumin, age, gender, or body mass index and polymorphisms of the most relevant genes encoding cytochrome-P450 enzymes, CYP3A4 and CYP3A5 [13]. However, it has been shown that a simple measurement such as the concentration dose ratio of tacrolimus (C/D) can predict risk to develop tacrolimus side effects. Some studies support that fast metabolizers of tacrolimus are more prone to show nephrotoxicity and polyoma BK nephropathy [14,15].

Clinical monitoring of renal transplants relies mainly on serial determinations of serum creatinine and proteinuria, therapeutic drug monitoring of treatments with a narrow therapeutic window (e.g., TAC-C_0_), determination of HLA antibodies by Luminex technology, and monitoring of the viral load of cytomegalovirus and polyoma BK virus. During the last decades, some centers have incorporated surveillance biopsies performed at different time points to detect the presence of subclinical inflammation and progression of renal scarring [16]. It has been proposed that in patients treated with tacrolimus and MMF, lower tacrolimus trough levels are associated with subclinical inflammation [17] and the progression of interstitial fibrosis and tubular atrophy (IF/TA) [18]. Similarly, a higher intra-patient variability of TAC-C_0_ has also been associated with a faster progression of IF/TA [19,20]. Finally, it has been shown that high tacrolimus clearance is a risk factor for the development of IF/TA in a large cohort of patients monitored by surveillance biopsies [21].

In this study, we explore the association between histological lesions and different surrogates of tacrolimus exposure/metabolism in a cohort of low immunological risk renal transplants treated with prolonged-release TAC (PR-TAC), MMF, and steroids monitored by paired surveillance biopsies. To analyze TAC exposure, we employed TAC-C0 at the time of each biopsy and assessed the intra-patient variability of TAC trough levels and time in/above/below the therapeutic range during follow up. To analyze TAC metabolism, we employed the tacrolimus C/D.

## 2. Patients and Methods

### 2.1. Patients

We conducted a prospective, longitudinal, observational study in renal transplants performed at our Renal Transplant Unit since January 2012 until December 2018. All living and donor deceased single kidney transplants performed in adult patients were considered. A first surveillance biopsy at 3–5 months after transplantation was performed in patients fulfilling the following criteria: (a) serum creatinine lower than 2 mg/dL; (b) stable renal function defined as a variability of serum creatinine lower than 15% between the determination at the time of biopsy and the previous one; (c) urinary protein creatinine ratio lower than 1 g/g; (d) non-use of oral anticoagulants; (e) non-technical difficulties to perform a renal biopsy (e.g., patients with large abdominal obesity, patients with large perirenal hematomas or patients with an idiomatic barrier were not considered) and (f) written informed consent. A second biopsy was performed in all patients at 12–18 months regardless of renal function and proteinuria.

This protocol was approved by the Ethics Committee of our center (PR/AG 104/2011), was performed in accordance with the Declaration of Helsinki, and is consistent with the Principles of the Declaration of Istanbul on Organ Trafficking and Transplant Tourism.

For the present study, we considered low immunological risk renal transplants, which are defined as the absence of donor-specific HLA antibodies at the time of transplant or having received a desensitization treatment before transplant in the case of living donors treated with prolonged-release tacrolimus (PR-TAC), mycophenolate mofetil (MMF) or enteric-coated mycophenolic acid (EC-MPA) and steroids during follow up. Patients treated with mTOR inhibitors (sirolimus or everolimus) from the day of transplant or switched to these drugs during follow up were also not considered.

### 2.2. Biopsies

Renal biopsies were performed under ultrasound guidance by trained radiologists with a 16-gauge automated needle. Three cores of tissue were obtained: one was processed for optical microscopy; one was embedded in OCT for immunofluorescence; and the other one was stored in RNA easy for molecular studies.

For optical microscopy, biopsies were embedded in formalin, paraffin-fixed, and 2–4 μm thick sections were stained with hematoxylin-eosin, periodic acid Schiff, Masson’s trichrome and silver methenamine. Sample adequacy and histological lesions were evaluated according to the last update of the Banff criteria by the renal pathologists [22].

Interstitial inflammation in surveillance biopsies was defined as i-score ≥ 1, while biopsies without interstitial infiltrates (i-score = 0) were classified as no inflammation. IF/TA score (ci + ct) was calculated for each biopsy and progression of IF/TA between biopsies was defined as a difference in ci + ct score between the second and first surveillance biopsy ≥ 1. Arteriolar hyalinosis progression was defined as a difference of ah-score between the second and first surveillance biopsy ≥ 1.

Immunofluorescence studies were performed in 3-μm cryostat sections stained with FITC-conjugated anti-human IgG, IgA, IgM, C3, κ and λ light chain. C4d was stained with indirect immunofluorescence with a monoclonal antibody (Quidel, San Diego, CA, USA), and its deposition in peritubular capillaries was graded according to the Banff criteria. All biopsies were stained with an anti-SV40 antibody to discard BK polyomavirus nephropathy.

### 2.3. Immunosuppression

Standard immunosuppression included the use of induction therapy for all renal transplants. Recipients of a first renal transplant with a calculated panel reactivity antibodies (cPRA) <50% received 20 mg of Basiliximab (Simulect^®^; Novartis, Basel, Switzerland) at days 0 and 4. Patients with previous transplants and/or with positive non-DSA anti-HLA antibodies with a cPRA ≥ 50% and/or receiving grafts from a deceased donor after cardiac death were treated with three to five doses of rabbit anti-thymocyte globulin (Thymoglobulin^®^; Sanofi-Aventis, Paris, France) on alternate days to reach a total dose of 3–6 mg/kg.

For the present study, we considered patients receiving maintenance immunosuppression based on the combination of PR-TAC (Advagraf^®^; Astellas Pharma, Meppel, The Netherlands), MMF (Cellcept^®^; Roche Pharmaceuticals, Basel, Switzerland), or EC-MPA (Myfortic^®^; Novartis, Basel, Switzerland), and steroids at the time of both surveillance biopsies. All patients received MMF 1 g bid (or EC-MPA 0.72 g bid) during the first month and 500 mg bid thereafter (or EC-MPA 0.36 g bid). In cases of suspected clinical intolerance to MMF or EC-MPA, further reductions of doses were done. The day of transplant patients received 250–500 mg of methylprednisolone, 125 mg at day 1 and 20 mg of prednisone at day 2. Thereafter, prednisone dose was progressively tapered to reach a daily dose of 0.1 mg/kg at 3 months and maintained during follow-up.

### 2.4. Therapeutic Drug Monitoring (TDM)

Tacrolimus trough levels in whole blood (TAC-C_0_) were measured by CMIA immunoassay (Abbott Laboratories^®^; Abbott Park, IL, USA), and the intra-assay and inter-assay coefficient of variation was lower than 6%. Target TAC-C_0_ were 8–12 ng/mL during the first 3 months after transplant and 6–10 ng/mL thereafter.

For the present study, TAC-C_0_ monitoring done at the following time periods was considered: weekly during the first month, every two weeks during the second and third months, monthly from 4 to 6 months, and every 2 months from 6 to 12 months. At the time of biopsies additional samples were obtained. To analyze the relationship between tacrolimus exposure/metabolism and histological lesions, we analyzed the intra-patient variability of tacrolimus concentration, the time on therapeutic range, and the concentration dose ratio (Figure 1).

Coefficient of variability (CV). Intra-patient variability of tacrolimus trough levels was evaluated as the CV calculated according to: CV (%) = (SD/mean) ∗ 100. For the present study, we considered CV of TAC-C_0_ between the first week after transplant and the first protocol biopsy and between both biopsies. The mean number of TAC-C_0_ determinations for the first period was 7.8 ± 1.7 (range 6–11) and for the second period, it was 6.1 ± 0.8 (range 5–7).

Time in therapeutic range (TTR). The linear interpolation method according to Rosendaal [23] was used to calculate TTR as well as the time below or above the therapeutic range. Briefly, this method assumes that a linear relationship exists between each measured value and then assigns a specific value for each day between tests (Figure 1). According to our immunosuppression schedule, the TAC-C_0_ therapeutic range was defined as 8–12 ng/mL during the first three months and 6–10 ng/mL thereafter. We calculated for each patient the number of days in, above, and below the therapeutic range and expressed the result as the percentage time for each studied period (from the first week until the first biopsy and between both biopsies).

Tacrolimus concentration dose ratio (C/D). Tacrolimus dose was recorded at 3, 6, and 12 months as well as the day of biopsies. The mean tacrolimus concentration dose ratio (C/D; ng/mL/mg) was calculated as the mean of concentration dose ratio at each time point.

### 2.5. Clinical Variables

Demographic characteristics of donors and recipients as well as transplant-related variables were recorded. Anti-HLA antibodies at the time of transplant and at the time of each biopsy were determined by Luminex technology using the product Lifecodes LifeScreen Deluxe (Gen-Probe; San Diego, CA, USA), and IgG specificities were examined by single antigen beads testing with Lifecodes Luminex single antigen class I and class II kits. At the time of each biopsy, serum creatinine, TAC-C_0_, and tacrolimus and MMF dose were recorded. In patients receiving EC-MPA, equimolar doses to MMF were used (720 mg of EC-MPA is equivalent to 1000 mg of MMF). Cytomegalovirus (CMV) infection was managed according to the international criteria [24]. Briefly, valganciclovir prophylaxis during the first 3–6 months was employed in high-risk patients (seropositive donor to seronegative recipient and patients treated with ATG) and a pre-emptive strategy with CMV viremia monitoring at each visit for the remaining patients. Monitoring of polyoma virus BK infection was done by the determination at each visit of BK viruria or BK viremia for those with increasing viral load in urine (>10^7^ log). In patients with increasing BK viral load, reduction of MMF and tacrolimus dose and/or switch to low tacrolimus dose and mTOR inhibitors was done according to the attending physician.

### 2.6. Statistics

Variables were described as frequencies, median, and interquartile range or mean and standard deviation for categorical, non-normally distributed continuous variables and normally distributed continuous variables, respectively. To compare paired data (Fisher exact test, Wilcoxon T test, or paired *t*-test) and unpaired data (Fisher exact test, Mann–Whitney U test, and *t*-test) appropriate tests were employed. Logistic regression analysis was employed to analyze the associations between histological lesions and clinical and TDM data. For multivariate logistic regression analysis, those variables with a *p*-value < 0.20 in the univariate analysis were considered. All tests were two-tailed, and a *p*-value < 0.05 was considered significant. Statistical analysis was done with Stata 13.1 software package (Stata Corp LP, College Station, TX, USA).

## 3. Results

### 3.1. Patients

During the study period, 692 renal transplants were performed at our center. The flow chart of included patients in the present study is shown in Figure 2. Demographic data and transplant-related variables from the studied cohort are shown in Table 1. Clinical data at the time of both surveillance biopsies as well as TDM are shown in Table 2. Renal function remains stable between biopsies. According to our protocol, tacrolimus doses and TAC-C_0_ were lower in the second period. The CV of TAC-C_0_ was lower, and TTR was higher during the second period. The bioavailability of TAC was slightly higher (higher C/D ratio) during the second period, but this difference did not reach statistical significance.

There exists a correlation between the different TDM methods employed in this study (correlation matrix at the time of both biopsies are shown in Appendix A).

### 3.2. Biopsies

The prevalence of subclinical rejection was low in both surveillance biopsies. In the first biopsy, there was one single case of T cell-mediated rejection (TCMR) grade IIA (isolated v-lesion in a patient without tubule-interstitial inflammation or microvascular inflammation and without HLA antibodies). There were no cases reaching criteria for tubulo-interstitial TCMR (*i* ≥ 2 and *t* ≥ 2). Borderline changes suspicious of TCMR (*i* ≥ and *t* ≥ 1 but lower than i2t2) were observed in eight cases (9.4%) in the first biopsy and in nine cases in the second one (10.6%). Isolated inflammation (*i* ≥ 1 and *t* = 0) was observed in 11 (12.9%) and in five cases (5.9%) in the first and second biopsies. Isolated tubulitis (*i* = 0 and *t* ≥ 1) was observed in six (7.1%) and in 13 (15.3%) cases in the first and second biopsy, respectively. No tubule-interstitial inflammation (*i* = 0 and *t* = 0) was observed in 60 (70.6%) and 58 (68.2%) in the first and second biopsies.

In the first biopsy, there were 43 cases (50.6%) with IF/TA (*ci* ≥ 1 and *ct* ≥ 1) that was mild (*n* = 38), moderate (*n* = 4), and severe (*n* = 1). In the second biopsy there were 57 cases (67.1%) with mild (*n* = 41), moderate (*n* = 13), and severe (*n* = 3) IF/TA. Criteria for chronic TCMR were fulfilled in two cases (2.3%) in the first and in five cases (5.9%) in the second biopsy. The degree of inflammatory lesions in any renal compartment did not change between biopsies, although tubulitis tended to be higher in the second biopsy. By contrast, chronic lesions including IF/TA and arteriolar hyalinosis significantly increased between biopsies (Table 3).

### 3.3. Interstitial Inflammation and TDM

Patients displaying interstitial inflammation (i-score ≥ 1) in the first surveillance biopsy had lower TAC-C0 levels and higher CV-TAC until biopsy. There was no association between TTR or time above/below the therapeutic range and tubule-interstitial inflammation (Table 4).

By logistic regression analysis, the presence of inflammation in the first surveillance biopsy was associated with TAC-C_0_ levels (odds ratio (OR): 0.69 and 95% confidence interval (CI): 0.50–0.96; *p* = 0.027) while CV-TAC until biopsy was not included. In the second surveillance biopsy, the time below the therapeutic range and TAC-C_0_ were lower in patients with inflammation (Table 5).

By logistic regression analysis, the presence of inflammation in the second surveillance biopsy was associated with the time below the therapeutic range (OR: 1.05 and 95% CI: 1.01–1.10; *p* = 0.023), while TAC-C_0_ was not included. Interstitial inflammation in scarred areas (i-IFTA score) was not associated with surrogates of TAC exposure/metabolism (Appendix A).

### 3.4. IF/TA Progression and TDM

Progression of IF/TA was observed in 35 cases (41.2%). Univariate analysis showed that IF/TA progression was associated with the mean C/D ratio and IF/TA score in the first biopsy (Table 6 and Table 7). Multivariate logistic regression analysis showed that mean C/D (OR: 0.48; 95% CI: 0.25–0.92; *p* = 0.026) and IF/TA in the first biopsy (OR: 0.43, 95% CI: 0.24–0.77, *p* = 0.005) were associated with IF/TA progression.

### 3.5. Arteriolar Hyalinosis Progression and TDM

During follow-up, arteriolar hyalinosis progressed in 31 cases (36.5%). Clinical and TDM data were not different among patients with and without arteriolar hyalinosis progression.

## 4. Discussion

A relationship between progression of IF/TA, evaluated by means of sequential surveillance biopsies, and surrogates of TAC exposure or metabolism, such as low TAC trough levels [18], high TAC variability [19,20], and C/D ratio [21], have been described in separate studies. In the present study, all these parameters were evaluated in the same cohort of patients to further characterize their relative contribution to IF/TA progression. We did not observe any association between TAC trough levels, TAC variability or time below the therapeutic range, and progression of IF/TA; while low C/D ratio, a surrogate of fast metabolizers, was associated with IF/TA progression. To interpret these observations, it should be taken into consideration that in the present cohort, TAC trough levels were relatively high in comparison to other immunosuppressive schedules [25], and time below the therapeutic range was lower than in previous reported studies. Regarding the relationship between high TAC variability and IF/TA, this relationship has only been observed in patients with a high variability and long-time periods of TAC levels below the therapeutic range [10]. Thus, the lack of a relationship between surrogates of TAC exposure and progression of IF/TA in the present study is consistent with the above-mentioned studies.

Subclinical inflammation has been associated with IF/TA progression [26,27], an increased risk for clinical rejection [28], the appearance of de novo donor-specific antibodies, especially in patients with a high number of HLA mismatches [12,29], the development of chronic antibody-mediated rejection [30], and decreased graft survival [31]. However, in this study, we failed to show an association between interstitial inflammation and IF/TA progression. This result may be explained by the low incidence and severity of inflammation. Only in one out of 170 biopsies did we observe subclinical rejection in a patient with isolated v-lesion, which is a histological phenotype that does not represent true rejection in a significant proportion of cases [32,33]. Approximately, there were 10% of early and late biopsies that displayed borderline changes and an additional 13% and 6% of early and late biopsies showed mild inflammation without tubulitis. Accordingly, in approximately 80% of cases, there was no interstitial inflammation. The relatively low incidence and severity of subclinical inflammation may be ascribed to the relatively high TAC trough levels that remained most of the time within the therapeutic range and to the low immunological risk of these set of patients [3,17,34]. We arbitrarily employed an i-score ≥ 1 as the threshold to distinguish between inflammation and no inflammation. This decision was based on previous observations showing that an i-score ≥ 1 is associated with decreased graft survival, while isolated tubulitis has a minor and controversial influence on outcome [35,36,37]. Despite the low degree of inflammation, TAC levels were significantly higher in early biopsies without inflammation, and the time below the therapeutic range was longer in late biopsies, suggesting that even in low immunological risk patients receiving a high TAC schedule, interstitial inflammation in healthy interstitial areas is modulated by TAC exposure. This observation constitutes an argument to sustain that subclinical inflammation in healthy areas may represent the balance between alloimmune response and immunosuppression. On the contrary, we did not observe any association between inflammation in scarred areas and TAC exposure or metabolism. The relationship between immunosuppressive treatment and inflammation in scarred tissue (i-IFTA) has been evaluated in few studies. In the study conducted by Lefaucheur et al. [38], the withdrawal of steroids, MMF, or calcineurin inhibitors at 6 months was associated with a higher risk of i-IFTA at 1 year. Similarly, in the study conducted by Nankivell et al. [39], tacrolimus was associated with a lower risk of i-IFTA than cyclosporine. In our study, all patients were treated with tacrolimus, MMF, and steroids until the second biopsy. Thus, in patients receiving a power immunosuppression, there was no association between i-IFTA and tacrolimus exposure, suggesting that other non-controlled factors in the present study contribute to the development of this lesion. Interstitial fibrosis, with independence of its trigger, is frequently associated with mononuclear cell infiltration [40,41,42]. In studies comparing gene expression in for-cause renal allograft biopsies with inflammation in scarred and unscarred areas, it was observed that inflammation in unscarred areas correlated with transcripts associated with cytotoxic T cells, while inflammation in scarred areas correlated with B cell, plasma cell, mast cell, and injury-repair transcripts [43,44]. This difference may be explained by the selective effect of tacrolimus on activated T cells [45,46]. Our data raise the question of whether the presence of inflammation in healthy kidney areas may facilitated TAC treatment adjustment to patient’s needs, following a personalized medicine approach. In the other hand, these data argue against the utility of inflammation in scarred areas to adjust TAC dose.

In the present study, there was an association between low C/D ratio, a surrogate of TAC metabolism, and the progression of IF/TA between both biopsies. Patients with a low C/D ratio represent fast metabolizers, since they need a high dose to reach the therapeutic levels. It has been previously described that a low C/D ratio is associated with poorer allograft function and a higher incidence of BK nephropathy and decreased allograft survival [15,47,48,49]. The pharmacokinetic curve in fast metabolizers is characterized by a higher peak TAC concentration (Cmax) in comparison with low metabolizers to reach a similar TAC-C0 [50,51]. Thus, it has been proposed that the progression of IF/TA in these patients can reflect TAC nephrotoxicity. In fact, according to our data, tacrolimus metabolism and not tacrolimus exposure contributes to IF/TA progression. The progression of IF/TA was also associated with the severity of IF/TA in the first biopsy, but this is an expected result that depends on the definition of ci and ct-scores according to the Banff criteria. Scoring for ci and ct is done according to the extension of interstitial fibrosis and tubular atrophy in the available tissue cortex as: ≤5% (ci = 0 and ct = 0); 6–25% (ci = 1 and ct = 1); 26–50% (ci = 2 and ct = 2); and >50% (ci = 3 and ct = 3). This kind of classification implies that patients without IF/TA in the first biopsy (ci and ct ≤ 5%) will have a higher risk of progression than patients with mild IF/TA (ci and ct of 6–25%) considering that the range of this last category is wider than the former. This association between IF/TA in the first biopsy and the risk of progression in biopsies performed later has been described in previous studies with paired surveillance biopsies [52].

Our study has some limitations. Preimplantation biopsies were not available, and we were not able to characterize IFTA progression from the donor to the first surveillance biopsy. Additionally, a 24-h pharmacokinetic study was not done to evaluate whether fast metabolizers (lower C/D ratio) have a higher tacrolimus Cmax than poor metabolizers (higher C/D ratio).

## 5. Conclusions

In conclusion, low C/D ratio is associated with IF/TA progression but not TAC trough levels, TAC variability, or time below the therapeutic range. Thus, TAC nephrotoxicity may contribute to fibrosis progression in well immunosuppressed patients. Additionally, we confirm that high TAC levels decrease inflammation in healthy kidney areas but not in scarred areas, pointing out that inflammation in scarred areas is not responsive to TAC exposure.

## Figures and Tables

**Figure 1 jcm-10-00141-f001:**
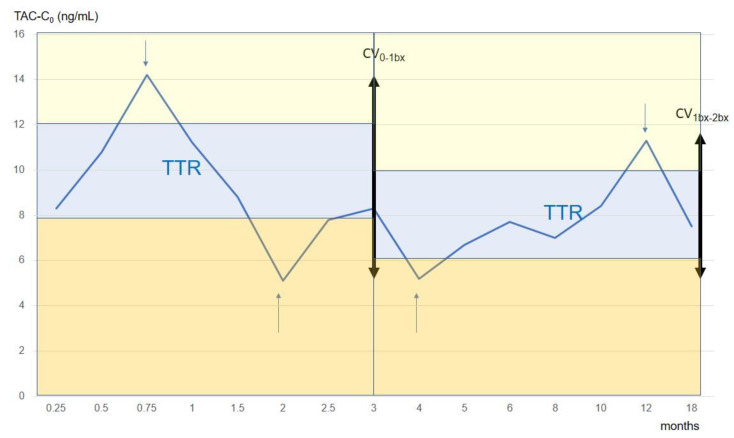
Evolution of tacrolimus trough levels during follow up in a patient from our cohort. TTR (blue area), time in the therapeutic range; times above (yellow area) and below (orange area) the therapeutic range are also shown. CV 0-1bx, coefficient of variability of tacrolimus between transplant and the first surveillance biopsy (29.3%); CV 1bx-2bx, coefficient of variability of tacrolimus between the first and the second surveillance biopsies (24.5%). In the *x*-axis, time of follow up (months) is presented, while in the *y*-axis, tacrolimus trough levels (ng/mL) are depicted. Arrows indicated determinations above and below the therapeutic range.

**Figure 2 jcm-10-00141-f002:**
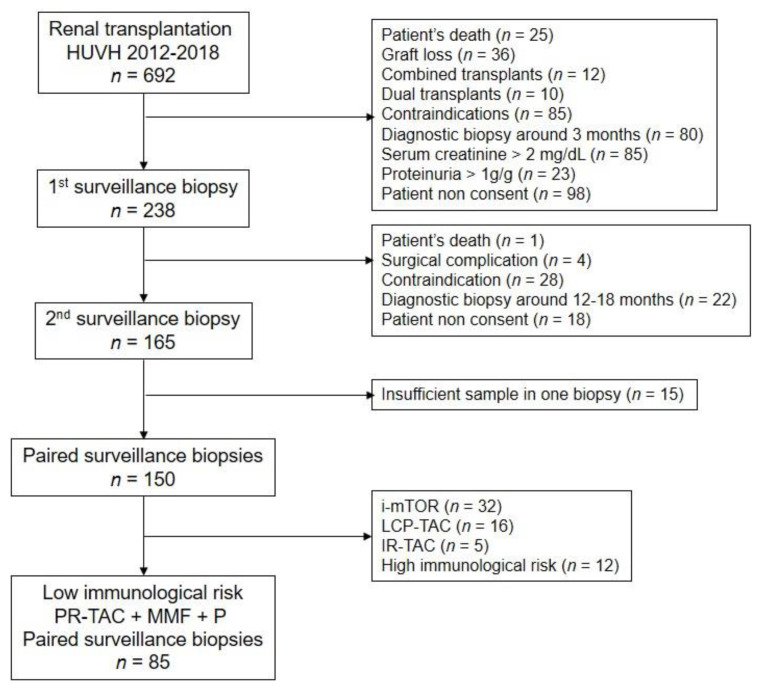
Flow chart of included patients. PR-TAC, prolonged-release tacrolimus; MMF, mycophenolate mofetil; P, prednisone; LCP-TAC, extended-release tacrolimus; IR-TAC, immediate-release tacrolimus; i-mTOR, inhibitors of mammalian target of rapamycin. Contraindications for the first surveillance biopsy include treatment with oral anticoagulants (*n* = 33), large abdominal obesity (*n* = 26), technical difficulties due to perirenal hematoma or lymphocele (*n* = 9), idiomatic barrier (*n* = 13), and horseshoe kidneys (*n* = 4). Indications for the use of mTOR inhibitors in this cohort was as follows: inclusion in a clinical trial containing i-mTOR de novo or early conversion (*n* = 16), polyoma BK viremia during follow up (*n* = 6), CMV viremia after prophylaxis in high-risk recipients (*n* = 8) and skin cancer (*n* = 2). High immunological risk patients were defined as those with HLA donor-specific antibodies at the time of transplant (*n* = 7) or receiving a desensitization treatment before transplant (*n* = 5).

**Table 1 jcm-10-00141-t001:** Donor and recipient characteristics as well as transplant related-variables from the studied cohort.

Variable	*n* = 85
Donor type (SCD/ECD/LD)	42 (49.4%)/30 (35.3%)/13 (15.3%)
Donor age (years)	52 ± 15
Donor gender (m/f)	50 (58.8%)/35 (41.2%)
Recipient age (years)	53 ± 13
Recipient gender (m/f)	66 (77.6%)/19 (22.4%)
First transplant/re-transplant	76 (89.4%)/9 (10.6%)
Primary renal disease (GN/ADPKD/diabetes/others/unknown)	21 (24.7%)/22 (25.9%)/8 (9.4%)/12 (14.1%)/22 (25.9%)
Class I HLA mismatch (A + B)	2.6 ± 0.9
Class II HLA mismatch (DR)	0.9 ± 0.7
Induction (Basiliximab/ATG)	52 (61.2%)/33 (38.8%)
DGF (no/yes)	80 (94%)/5 (6.0%)
T cell-mediated rejection (no/yes)	81 (95.3%)/4 (4.7%)
CMV infection (no/viremia/disease)	71 (83.5%)/10 (11.8%)/4 (4.7%)
Polyoma BK virus infection (no/viremia/nephropathy)	79 (92.9%)/6 (7.1%)/0 (0%)
Post-transplant diabetes mellitus (no/yes)	61 (79.2%)/16 (20.8%)

SCD, standard criteria deceased donor; ECD, expanded criteria deceased donor; LD, living donor; GN, glomerulonephritis; ADPKD, autosomal dominant polycystic kidney disease; DGF, delayed graft function; CMV, cytomegalovirus.

**Table 2 jcm-10-00141-t002:** Clinical data at the time of the first and second surveillance biopsies of the studied cohort (*n* = 85).

Variable	First Biopsy	Second Biopsy	*p*-Value
Time of biopsy (months)	4.2 ± 1.9	17.3 ± 3.6	n.a.
Serum creatinine (mg/dL)	1.31 ± 0.32	1.27 ± 0.28	0.106
eGFR (mL/min/1.73 sqm)	61.8 ± 17.7	63.4 ± 17.1	0.220
Urine P/C ratio (mg/g)	260 ± 170	320 ± 370	0.057
Tacrolimus dose (mg/day)	6.6 ± 3.9	4.9 ± 3.1	0.001
MMF dose (mg/kg/day)	13.4 ± 3.4	12.9 ± 3.3	0.096
TAC-C0 (ng/mL)	9.6 ± 2.4	8.5 ± 2.3	0.002
CV of TAC-C0 (%)	31 ± 13	20 ± 14	0.001
Time in TR(%)	55 ± 24	70 ± 25	0.001
Time above TR (%)	35 ± 25	26 ± 29	0.066
Time below TR (%)	10 ± 13	4 ± 11	0.004
C/D (ng/mL/mg)	2.00 ± 1.42	2.19 ± 1.02	0.119
De novo DSA (%)	0	0	n.a.

eGFR, estimated glomerular filtration rate by MDRD-4 formula; urine P/C ratio, urine protein creatinine ratio; TAC-C0, tacrolimus trough levels; CV of TAC-C0, coefficient of variability of tacrolimus trough levels from the first week until the first biopsy and between the first and the second biopsies; TR, percentage of time in/above/below the therapeutic range until the first biopsy and between the first and the second biopsies; C/D, concentration dose ratio of tacrolimus; DSA, donor specific antibodies.

**Table 3 jcm-10-00141-t003:** Banff scores in the first and second surveillance biopsies (*n* = 85).

Variable	First Biopsy	Second Biopsy	*p*-Value
Glomerular sections (*n*)	13 ± 8	13 ± 8	0.881
Global glomerulosclerosis (%)	7 ± 10	8 ± 11	0.541
Glomerulitis (g)	0.12 ± 0.36	0.15 ± 0.45	0.580
Interstitial infiltrate (i)	0.27 ± 0.54	0.17 ± 0.37	0.118
Tubulitis (t)	0.18 ± 0.42	0.31 ± 0.58	0.086
Endothelialitis (v)	0.01 ± 0.11	0	0.320
Peritubular capillaritis (ptc)	0.13 ± 0.37	0.18 ± 0.47	0.413
Arteriolar hyalinosis (ah)	0.38 ± 0.64	0.65 ± 0.79	0.008
Transplant glomerulopathy (cg)	0	0.04 ± 0.24	0.183
Interstitial fibrosis (ci)	0.73 ± 0.77	1.00 ± 0.84	0.002
Tubular atrophy (ct)	0.76 ± 0.59	1.08 ± 0.73	0.001
Vascular intimal thickening (cv)	0.62 ± 0.91	0.67 ± 0.85	0.636
Mesangial matrix increase (mm)	0.02 ± 0.15	0.01 ± 0.11	0.567
i-IFTA	1.28 ± 1.17	1.39 ± 1.21	0.494
t-IFTA	0.36 ± 0.55	0.51 ± 0.67	0.128

i-IFTA. Interstitial inflammation in areas of interstitial fibrosis and tubular atrophy; t-IFTA, tubulitis in areas of interstitial fibrosis and tubular atrophy.

**Table 4 jcm-10-00141-t004:** Clinical and therapeutic drug monitoring data in patients with (i-score ≥1) and without (i-score = 0) interstitial inflammation in the first surveillance biopsy.

Variable	No Inflammation	Inflammation	*p*-Value
	(*n* = 66)	(*n* = 19)	
Time of biopsy (months)	4.3 ± 1.5	3.7 ± 2.9	0.167
Serum creatinine (mg/dL)	1.34 ± 0.33	1.20 ± 0.26	0.106
eGFR (mL/min/1.73 sqm)	61 ± 17	65 ± 19	0.369
Urine P/C ratio (mg/g)	270 ± 180	220 ± 130	0.252
Tacrolimus dose (mg/day)	6.7 ± 3.9	5.4 ± 3.5	0.184
MMF dose (mg/kg/day)	13.2 ± 3.3	13.5 ± 3.8	0.795
TAC-C0 (ng/mL)	10.0 ± 2.4	8.3 ± 2.2	0.007
CV of TAC-C0 (%)	29 ± 12	37 ± 15	0.030
Time in TR (%)	56 ± 26	55 ± 21	0.889
Time above TR (%)	36 ± 26	33 ± 21	0.642
Time below TR (%)	8± 13	12 ± 14	0.246
C/D (ng/mL/mg)	1.88 ± 1.18	2.45 ± 2.03	0.124

eGFR, estimated glomerular filtration rate by MDRD-4 formula; urine P/C ratio, urine protein creatinine ratio; TAC-C0, tacrolimus trough levels; CV of TAC-C0, coefficient of variability of tacrolimus trough levels from the first week until the first biopsy; TR, percentage of time in/above/below the therapeutic range from the first week until the first biopsy; C/D, concentration dose ratio of tacrolimus.

**Table 5 jcm-10-00141-t005:** Clinical and therapeutic drug monitoring data in patients with (i-score ≥ 1) and without (i-score = 0) interstitial inflammation in the second surveillance biopsy.

Variable	No Inflammation	Inflammation	*p*-Value
	(*n* = 71)	(*n* = 14)	
Time of biopsy (months)	17.4 ± 3.8	16.4 ± 2.5	0.380
Serum creatinine (mg/dL)	1.26 ± 0.26	1.33 ± 0.37	0.405
eGFR (mL/min/1.73 sqm)	64.4 ± 17.4	58.2 ± 15.2	0.217
Urine P/C ratio (mg/g)	301 ± 348	423 ± 494	0.270
Tacrolimus dose (mg/day)	5.0 ± 3.3	4.3 ± 2.1	0.426
MMF dose (mg/kg/day)	12.7 ± 3.5	13.1 ± 1.7	0.691
TAC-C0 (ng/mL)	8.7 ± 2.3	7.4 ± 1.7	0.059
CV of TAC-C0 (%)	20 ± 15	19 ± 7	0.826
Time in TR (%)	70 ± 27	74 ± 19	0.534
Time above TR (%)	27 ± 27	14 ± 21	0.076
Time below TR (%)	3 ± 10	12 ± 15	0.005
C/D (ng/mL/mg)	2.22 ± 1.04	2.04 ± 0.82	0.557

eGFR, estimated glomerular filtration rate by MDRD-4 formula; urine P/C ratio, urine protein creatinine ratio; TAC-C0, tacrolimus trough levels; CV of TAC-C0, coefficient of variability of tacrolimus trough levels from the first until the second biopsy; TR, percentage of time in/above/below the therapeutic range from the first until the second biopsy; C/D, concentration dose ratio of tacrolimus.

**Table 6 jcm-10-00141-t006:** Clinical and therapeutic drug monitoring data in patients with and without interstitial fibrosis and tubular atrophy progression between biopsies.

Variable	No Progression	IFTA Progression	*p*-Value
	(*n* = 50)	(*n* = 35)	
Donor age (years)	54 ± 16	50 ± 13	0.285
Donor type (DD/LD)	43 (86%)/7 (14%)	29 (82.8%)/6 (17.2%)	0.382
Recipient age (years)	54 ± 13	50 ± 14	0.184
1st transplant/re-transplant	45 (90%)/5 (10%)	31 (88.6%)/4 (11.4%)	0.551
HLA-DR mismatch	1.0 ± 0.7	0.9 ± 0.7	0.856
Cold ischemia time (hours)	15 ± 7	14 ± 8	0.850
Induction (IL2-RA/ATG)	29 (58%)/21 (42%)	22 (62.9%)/13 (37.1%)	0.558
DGF (no/yes)	48 (96%)/2 (4%)	32 (91.4%)/3 (8.6%)	0.399
T-cell mediated rejection (no/yes)	47 (94%)/3 (6%)	34 (97%)/1 (3%)	0.640
Time of biopsy (months)	17.7 ± 4.1	16.6 ± 2.7	0.156
Serum creatinine (mg/dL)	1.25 ± 0.29	1.29 ± 0.27	0.582
eGFR (mL/min/1.73 sqm)	64 ± 18	63 ± 16	0.881
Urine P/C ratio (mg/g)	310 ± 340	350 ± 425	0.622
Tacrolimus dose 1st bx (mg/day)	5.8 ± 3.8	7.4 ± 3.8	0.058
Tacrolimus dose 2nd bx (mg/day)	4.3 ± 2.9	5.8 ± 3.2	0.023
MMF dose (mg/kg/day)	12.9 ± 3.2	12.7 ± 3.4	0.836
TAC-C0 1st biopsy (ng/mL)	9.8 ± 2.7	9.4 ± 1.9	0.515
TAC-C0 2nd biopsy (ng/mL)	8.1 ± 1.7	9.1 ± 2.8	0.053
CV of TAC-C0 until first biopsy (%)	32 ± 13	30 ± 13	0.503
CV of TAC-C_0_ between biopsies (%)	21 ± 18	18 ± 7	0.305
Time in TR between biopsies (%)	73 ± 25	67 ± 27	0.353
Time above TR between biopsies (%)	22 ± 25	29 ± 29	0.318
Time below TR between biopsies (%)	5 ± 12	4 ± 9	0.798
Mean C/D (ng/mL/mg)	2.3 ± 1.3	1.7 ± 0.7	0.019

DD, deceased donor; LD, living donor; DGF, delayed graft function; eGFR, estimated glomerular filtration rate by MDRD-4 formula; urine P/C ratio, urine protein creatinine ratio; TAC-C0, tacrolimus trough levels; CV of TAC-C0, coefficient of variability of tacrolimus trough levels; TR, percentage of time in/above/below the therapeutic range; C/D, concentration dose ratio of tacrolimus.

**Table 7 jcm-10-00141-t007:** Banff scores in the first and second surveillance biopsies in patients with and without IF/TA progression between biopsies.

Variable	No Progression	IFTA Progression	*p*-Value
	(*n* = 50)	(*n* = 35)	
g+ptc score 1st biopsy	0.32 ± 0.74	0.14 ± 0.43	0.207
i+t score 1st biopsy	0.56 ± 0.91	0.29 ± 0.62	0.125
i-IFTA score 1st biopsy	1.43 ± 1.19	1.03 ± 1.12	0.124
ci+ct score 1st biopsy	1.90 ± 1.27	0.89 ± 0.93	0.001
ah-score 1st biopsy	0.40 ± 0.61	0.34 ± 0.69	0.686
g+ptc score 2nd biopsy	0.24± 0.56	0.46 ± 0.95	0.188
i+t score 2nd biopsy	0.48 ± 0.89	0.46 ± 0.70	0.899
ci+ct score 2nd biopsy	1.60 ± 1.28	2.71 ± 1.64	0.001
i-IFTA score 2nd biopsy	1.20 ± 1.27	1.69 ± 1.08	0.073
ah-score 2nd biopsy	0.69 ± 0.77	0.60 ± 0.85	0.598

g, glomerulitis; ptc, peritubular capillaritis; i, interstitial inflammation in non-scarred cortex; t, tubulitis; ci, interstitial fibrosis; ct, tubular atrophy; i-total, interstitial inflammation in the whole cortex; i-IFTA, interstitial inflammation in scarred cortex; ah, arteriolar hyalinosis.

## Data Availability

The data presented in this study are available on request from the corresponding author.

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
