# Peer review of "Progression of Interstitial Fibrosis and Tubular Atrophy in Low Immunological Risk Renal Transplants Monitored by Sequential Surveillance Biopsies: The Influence of TAC Exposure and Metabolism"

_jcm, 2021, doi:10.3390/jcm10010141_

Round 1

Reviewer 1 Report

In this study, Chamoun and co-authors analyzed the association between different surrogates for Tacrolimus exposure (C0, coefficient of variation, time in therapeutic range and concentration dose ratio C/D) to histological lesions especially the progression of IF/TA on M3 and M12 protocol biopsies in 85 low immunological kidney transplant recipients. They found that C0 concentration is associated with inflammation in the first biopsy and that time under therapeutic range to inflammation in the second biopsy. Progression of IFTA which was found in 35 patients and was associated by multivariate logistic regression to a low C/D ratio which can reflect TAC nephrotoxicity. Interstingly  i-IFTA was not associated with TAC exposure.

This is an interesting study showing that underexposure to Tacrolimus may be associated with inflammation in non-scared areas in protocol biopsies in a cohort of low immunological risk transplant recipients, but not with i-IFTA, and also that C/D Tacrolimus ratio but not C0 is associated to IF/TA progression.

I have several minor concerns regarding this study:

  • In table 1, 4 patients experienced TCMR. However, in the results, it is said that there was only one case of TCMR grade IIA (isolated v lesion) and 2 chronic TCMR in the first biopsy and 5 in the second? It would clearer to write in Table 1 the numbers and delay of Borderline, active TCMR, chronic active TCMR.
  • Was there any case of ABMR?
  • Did the center perform Time zero biopsies in order to analyze IFTA progression also between day 0 and month 3? Indeed, patients with no IFTA progression had more IFTA at 3 months. Was it a fibrosis inherited from the donor?
  • The authors state that there is a correlation between the different TDM methods and this is showed in supplementary Table 1 and 2, however C/D ratio is not well correlated to TAC C0 and shows the best association with progression of IFTA. This could suggest that this parameter should be more often taken in consideration for TAC nephrotoxicity. This could be more discussed.
  • There was no association between i-IFTA and TAC exposure in this study. However, it was showed that underimmunosuppression was associated with i-IFTA at one year in 2 studies which should be discussed in the discussion section (Lefaucheur AJT 2018 Feb;18(2):377-390; nankivell AJT 2018 Feb;18(2):364-376)
  • The last sentence in the discussion section to explain why IF/TA progression is associated with a lower IFTA in the first biopsy is not very clear, and may not be necessary. It is indeed interesting that patient without progression have a higher IFTA at 3 months (was it already present at the time of transplant?) because high IF/TA at 3 months is often a criteria to guide the decrease of immunosuppression, whereas according to the results of this study, it should be also considered in patient with a lower IFTA at 3 months, especially with a high C/D ratio.

Other remarks:

Give also the percentage in table 1 and 6

Table 2: add the unit for “time in, above and under TR

In the title of Table 2, there is a typo error (“at the time” twice)

Author Response

Response to Reviewer Comments 1

  • In table 1, 4 patients experienced TCMR. However, in the results, it is said that there was only one case of TCMR grade IIA (isolated v lesion) and 2 chronic TCMR in the first biopsy and 5 in the second? It would clearer to write in Table 1 the numbers and delay of Borderline, active TCMR, chronic active TCMR.In table 1 we showed the baseline characteristics of our cohort of patients before performing the first surveillance biopsy. From our cohort, there were 4 out of 85 patients in whom a biopsy for indication was done during the initial 3 months after transplantation and were diagnosed of T cell-mediated rejection (and accordingly treated). Thereafter, we showed the results of surveillance biopsies. In these set of surveillance biopsies there was only one case of TCMR.
    •  
    •  
    • Was there any case of ABMR?  
    • In our set of low immunological risk patients there was no episodes of ABMR. All patients displaying DSA at the time of transplant or receiving a desensitization protocol before transplant were not included in the present study. From our set of patients anyone developed de novo DSA during the time of follow up.
  • Did the center perform Time zero biopsies in order to analyze IFTA progression also between day 0 and month 3? Indeed, patients with no IFTA progression had more IFTA at 3 months. Was it a fibrosis inherited from the donor?We agree with the reviewer than the analysis of IF/TA in pre-implantation biopsies will allow to properly characterize its evolution after transplantation (we have done such a work some years ago with a small sample of patients: Moreso F et al. Am J Transplant 2001; 1: 82-88). However, we did not perform routine pre-implantation biopsies for all renal transplants. For living donors and standard criteria donors (age < 60 years) pre-implantation biopsies were not perform. Thus, from our cohort only in 35% of transplants performed with an ECD donor a pre-implantation biopsy is available. Thus, we cannot perform the analysis suggested by the reviewer and we have added this point in a paragraph of limitations of our study.  
    • We agree with the reviewer that recipients without IFTA progression between biopsies displayed a higher ci+ct score in the 1st surveillance biopsy and, possibly, this reflects donor-derived fibrosis. In fact, donor age tended to be higher in patients without IFTA progression (54 vs. 50 years, table 6). However, it should be stressed that the scoring for ci and ct according to the Banff schema is done according to the extension of interstitial fibrosis and tubular atrophy in the available tissue cortex as: ≤ 5% (ci=0 and ct=0); 6-25% (ci=1 and ct=1); 26-50% (ci=2 and ct=2); > 50% (ci=3 and ct=3). This kind of classification implies that patients without IFTA in the first biopsy (ci and ct ≤ 5%) will have a higher risk of progression than patients with mild IFTA (ci and ct of 6-25%). This association between IFTA in the first biopsy and the risk of progression was extensively analysed in a publication of our group analysing a large cohort (n=155) of patients with paired surveillance biopsies (Seron D, Moreso F et al. Kidney Int 2002; ref 52 of the present version).
    •  
  • The authors state that there is a correlation between the different TDM methods and this is showed in supplementary Table 1 and 2, however C/D ratio is not well correlated to TAC C0 and shows the best association with progression of IFTA. This could suggest that this parameter should be more often taken in consideration for TAC nephrotoxicity. This could be more discussed. 
    • C/D ratio was not associated with TAC-C0 in the first biopsy and shows a mild association with TAC-C0 in the second one (rho=0.236). We agree with the reviewer that according to our data, C/D ratio is a better surrogate of TAC nephrotoxicity than TAC-C0. We further discussed this result.
  • There was no association between i-IFTA and TAC exposure in this study. However, it was showed that underimmunosuppression was associated with i-IFTA at one year in 2 studies which should be discussed in the discussion section (Lefaucheur AJT 2018 Feb;18(2):377-390; Nankivell AJT 2018 Feb;18(2):364-376).We agree with the reviewer that underimmunosuppression has been associated with i-IFTA in the above-mentioned studies (we added both studies to the references). In the study conducted by Lefaucheur et al withdrawal of with steroids, CNI or MMF at 6-months were associated with a higher risk of i-IFTA at 1-year. In our cohort all patients received this schedule until the second surveillance biopsy. In the study conducted by Nankivell et al, tacrolimus was associated with a lower risk of i-IFTA than cyclosporine. In our study all patients were treated with tacrolimus. Thus, in our cohort of patients receiving a power immunosuppression schedule we were not able to detect an association between i-IFTA and tacrolimus exposure suggesting than other non-controlled factors in the present study contributes to the development of this lesion. This point has been added to the discussion.
    •  
    •  
  • The last sentence in the discussion section to explain why IF/TA progression is associated with a lower IFTA in the first biopsy is not very clear, and may not be necessary. This sentence has been completely rewritten to clarify its meaning.
    •  
    •  
  • It is indeed interesting that patient without progression have a higher IFTA at 3 months (was it already present at the time of transplant?) because high IF/TA at 3 months is often a criterion to guide the decrease of immunosuppression, whereas according to the results of this study, it should be also considered in patient with a lower IFTA at 3 months, especially with a high C/D ratio. We agree with the reviewer comments. Our data showed that IFTA in the first biopsy and tacrolimus C/D ratio were independent predictors of IFTA progression between biopsies. This means, the risk to progress for fast metabolizers (low C/D ratio) is independent from the degree of IFTA at the time of the first biopsy. Thus, in patients with low IFTA, the C/D ratio should be taken into consideration since it can modulate the risk of progression.  
  • Other remarks:
  •  
  •  
  • Give also the percentage in table 1 and 6.  
    • We have added percentages in table 1 and 6.
  • Table 2: add the unit for “time in, above and under TR. 
    • We have added the unit, this is, percentage of the timeframe.
  • In the title of Table 2, there is a typo error (“at the time” twice)
  • This typo has been corrected. Thanks for your help.

Reviewer 2 Report

A study is well designed and interesting. Methods and results are clearly  described. In my opinion conclusion that “low C/D ratio is associated with IF/TA progression” can be made from study results, but we cannot conclude that “TAC nephrotoxicity may constitute the main driver of fibrosis progression” as deeper evaluation of tacrolimus pharmacokinetics was not done and there was no difference in Tac trough levels in IFTA progression group as compared to no progression in your study. From table 7 we can see that in IFTA progression group there is also increase in i+t score from 1st to 2nd biopsy - could it be some inflammation going on?

It is not clear for me why did you select serum creatinine lower than 2 mg/dL as inclusion criteria, also variability of serum creatinine 15, and not 20 or 30 %? Histological changes from 1st to 2nd biopsy could also be associated with tacrolimus kinetics in these patients.

Author Response

Response to Reviewer 2

  • The study is well designed and interesting. Methods and results are clearly described. In my opinion conclusion that “low C/D ratio is associated with IF/TA progression” can be made from study results, but we cannot conclude that “TAC nephrotoxicity may constitute the main driver of fibrosis progression” as deeper evaluation of tacrolimus pharmacokinetics was not done and there was no difference in Tac trough levels in IFTA progression group as compared to no progression in your study. From table 7 we can see that in IFTA progression group there is also increase in i+t score from 1st to 2nd biopsy - could it be some inflammation going on?
  • Thanks for the general comment of the reviewer. We agree with the reviewer that the conclusion: “TAC nephrotoxicity may constitute the main driver of fibrosis progression” is an overinterpretation and we have tempered this sentence in the abstract and discussion as: “TAC nephrotoxicity may contribute to fibrosis progression”. We have added a paragraph of limitations and we included that a 24-hour pharmacokinetic study was not done to evaluate whether fast metabolizers (lower C/D ratio) have a higher tacrolimus Cmax than poor metabolizers (higher C/D ratio). In table 7, we described i+t score in patients with IFTA progression (0.29 ± 0.62 in the first biopsy and 0.46 ± 0.70 in the second one). This difference is not significantly different (p-value=0.28) and our data did not allow to support that in our set of patients receiving a power immunosuppression, subclinical inflammation significantly contributes to IFTA progression.
    •  
  • It is not clear for me why did you select serum creatinine lower than 2 mg/dL as inclusion criteria, also variability of serum creatinine 15, and not 20 or 30 %? Histological changes from 1st to 2nd biopsy could also be associated with tacrolimus kinetics in these patients.
  • Criteria employed by different centers to perform surveillance biopsies are not homogenous. There are some centers who perform surveillance biopsies at predefined time points regardless of any clinical data. In this case, patients with acute or chronic dysfunction will be included and, thus, histological changes will be more severe (probably with more cases of acute rejection and borderline changes).   
  • When we start our surveillance biopsy program we decided “a priori” to include at the time of the first surveillance biopsy only well-functioning grafts (serum creatinine < 2 mg/dL with proteinuria < 1g/g) with stable function (variability of serum creatinine < 15%). We understand that this design has some limitations since it does not allow to monitor patients with suboptimal and/or unstable renal function. However, in our opinion, these patients will require a biopsy for cause to analyze histological changes and adjust immunosuppression accordingly.  

Reviewer 3 Report

General Comments

The authors assessed the surrogate marker of the exposure of TAC by paired surveillance biopsies using the relationship between TAC-C0, CV-TAC-C0, TTR, or C/D and Banff i-score, inflammation, or IF/TA progression. However, the way to show the results were not fully matured in the manuscript.

Major revision

  1. In the conclusion, the authors emphasized that TAC exposure is associated with inflammation in healthy kidney areas but not in scarred tissue. However, these things were just discussion and not showed clearly in the results. If the author would like to state this, the author needs to show the comparison of inflammation scores between the healthy and scarred area in the biopsied tissue.
  2. The author described the results of the multivariate logistic regression analysis. However, there is no data on multivariate analysis in the Tables. The authors should show them.
  3. There are Tables 6 and 7 in the results. However, there was no description of the results from Tables 6 and 7.

Minor revision

  1. It is necessary to display the horizontal axis and the vertical axis in Figure 1.
  2. In Table4 and 5, the “Time below TR (%) was described in the results as “ the time under the therapeutic range”.
  3. There are bold “p-value” in the Tables which could be “statistically significant”. However, the standard of the bold seems to be not defined consistently.

Author Response

Responses to Reviewer comments 3

General Comments

The authors assessed the surrogate marker of the exposure of TAC by paired surveillance biopsies using the relationship between TAC-C0, CV-TAC-C0, TTR, or C/D and Banff i-score, inflammation, or IF/TA progression. However, the way to show the results were not fully matured in the manuscript.

We appreciate the comment of the reviewer and we have tried to improve our manuscript after this revision considering the limitations underlined by this reviewer.

Major revision

  • In the conclusion, the authors emphasized that TAC exposure is associated with inflammation in healthy kidney areas but not in scarred tissue. However, these things were just discussion and not showed clearly in the results. If the author would like to state this, the author needs to show the comparison of inflammation scores between the healthy and scarred area in the biopsied tissue.

The association between inflammation in non-scarred tissue and TDM is shown in tables 4 and 5 (first and second surveillance biopsies). In the line 295 we described: “Interstitial inflammation in scarred areas (i-IFTA score) was not associated with surrogates of TAC exposure / metabolism (suppl. Tables 3 and 4)”. To shorten the number of tables in the paper we decided to include this tables as supplementary since there were no statistical differences among groups. We led the decision to include these tables in the main text to the editors.

  • The author described the results of the multivariate logistic regression analysis. However, there is no data on multivariate analysis in the Tables. The authors should show them.

In tables 4 to 7 we shown the results of the univariate analysis to be able to characterize the absolute difference between compared groups (no inflammation vs. inflammation in early and late biopsies as well as IF/TA progression). Later, we perform a logistic regression analysis to evaluate the association between clinical/TDM and inflammation in surveillance biopsies as well as IF/TA progression. It is possible to add a new table with the results of univariate and multivariate regression analysis but since only 1 or 2 variables are included in the multivariate models we decided to show the results (OR, 95% CI and p-value) into the body of the text.

  • There are Tables 6 and 7 in the results. However, there was no description of the results from Tables 6 and 7.

Line 298 states that: “Progression of IF/TA was observed in 35 cases (41.2%). Univariate analysis showed that IF/TA progression was associated with the mean C/D ratio and IF/TA score in the first biopsy (Tables 6 and 7)”. In both tables we compare clinical, TDM and histological variables in patients with IFTA progression and patients without progression.

Minor revision

  • It is necessary to display the horizontal axis and the vertical axis in Fig 1. 
  • We apologize for this mistake that has been corrected in the new version.
  • In Table 4 and 5, the “Time below TR (%) was described in the results as “the time under the therapeutic range”.
  • We have corrected it and we describe this variable as time below the therapeutic range.

  • There are bold “p-value” in the Tables which could be “statistically significant”. However, the standard of the bold seems to be not defined consistently.
  • We apologize for this mistake. Only p-values < 0.05 should be shown in bold format. We have corrected all p-values properly.

Round 2

Reviewer 3 Report

All the mentioned issues were appropriately responded to and corrected in this version of the manuscript. There were several minor revisions in terms of spelling in the text.

  1. line 261: in 5 cases (5,9%) should be "5.9%".
  2. line 347: Arteriolar hyalinosis progression ad TDM should be "and".

Author Response

We appreciate the comment of the reviewer and we have revised it.